# Plasmonic Biosensors: Review

**DOI:** 10.3390/biology11050621

**Published:** 2022-04-19

**Authors:** Mohga E. Hamza, Muhammad A. Othman, Mohamed A. Swillam

**Affiliations:** Nanophotonics Research Laboratory, Department of Physics, The American University in Cairo, Cairo 11835, Egypt; mohga.essam@aucegypt.edu (M.E.H.); muhammad.othman@aucegypt.edu (M.A.O.)

**Keywords:** biosensors, plasmonics, surface plasmon resonance, lab-on-chip

## Abstract

**Simple Summary:**

Plasmonic biosensors have extremely unique properties due to their high specificity and sensitivity in the detection of pathogens, making them of great interest to be used in various areas such as viral detection, diagnosing pollutants in the environment, and monitoring biomolecules in food. Plasmonic biosensors work by functionalizing a surface with an antibody which upon binding to an antigen results in variation in the angle of reflection which represents changes in the resonance conditions observed. Plasmonic biosensors are therefore thoroughly designed, fabricated, and optimized using explicitly selected plasmonic substrates. Hence, choosing the appropriate materials, structures, and functionality is of great importance to develop plasmonic biosensors highly specific for their application in real-life situations. Sensors based on surface plasmon resonance (SPR), localized SPR (LSPR), and Surface-enhanced Raman scattering (SERS) have been used over the years to monitor environmental biological molecules, ensure food safety, and detect up-to-date pathogens and viruses. This review article will discuss the mechanism of plasmonic biosensors, methods of controlling the efficiency, and their different designs when used in various applications.

**Abstract:**

Biosensors have globally been considered as biomedical diagnostic tools required in abundant areas including the development of diseases, detection of viruses, diagnosing ecological pollution, food monitoring, and a wide range of other diagnostic and therapeutic biomedical research. Recently, the broadly emerging and promising technique of plasmonic resonance has proven to provide label-free and highly sensitive real-time analysis when used in biosensing applications. In this review, a thorough discussion regarding the most recent techniques used in the design, fabrication, and characterization of plasmonic biosensors is conducted in addition to a comparison between those techniques with regard to their advantages and possible drawbacks when applied in different fields.

## 1. Introduction

Over the years, biosensors have been used as analytical tools that take a biological response as an input and translate it into an electrical signal. As per the International Union of Pure and Applied Chemistry (IUPAC), biosensors are self-contained integrated devices that can offer quantitative results that are thoroughly analyzed via biological recognitions or receptors in contact with a transducer [1]. Overall, biosensors are designed to have high specificity, selectivity, independence of physical restrictions like pH and temperature, and several other advantages, making them high in demand [2,3,4,5,6].

Biosensors were used in several applications including environmental evaluation, medical diagnosis, metabolic engineering, and food analysis [7,8]. As highly accurate analytical devices, biosensors recognize and scrutinize biological analytes such as enzymes, antibodies, aptamers, lectins, or nucleic acids via bioreceptors as shown in Figure 1a [1]. A transducer then converts the analyzed results into signals which get amplified and detected by a physiochemical detector to form an optical signal or a digitized output as shown in Figure 1b [9]. Fabricating a biosensor requires materials that are characterized depending on their mechanism based on two main categories. The first category includes biocatalytic groups involving enzymes via immobilization methods. The second category would include bio-affinity groups having antibodies and nucleic acids which could be natural or artificial, single-stranded or double-stranded nucleic acids, RNA/DNA hybrids, and anti-sense oligonucleotides [4].

Biosensors offer accurate analyte concentrations due to their direct and linear relationship to the intensity of the signal that requires detection. Hence, this helps in predicting the sensitivity of various biosensors. The most common pathogen detecting biosensors used are magnetic [10], colorimetric [11], electrochemical [12], lateral-flow [13], and optical biosensors [14]. Recently, research on optical biosensors, especially plasmonic and metamaterial-based biosensors, has grown massively due to their wide range of advantages including being versatile, highly-sensitive, reusable, affordable, label-free monitoring, ultra-low detection limits, and real-time sensing [15,16,17,18,19,20].

In this paper, we start by discussing the mechanism of plasmonic biosensors and their efficiency followed by a thorough description of the application of plasmonic biosensors as platforms for virus detection, particularly with SARS-CoV-2, environmental evaluators, and food monitoring and analysis. The next part will introduce the intervention of metamaterials in biosensors which provides them with conspicuously higher precision. Recent advances, limitations, future challenges, and opportunities in this field will then be presented.

### 1.1. Mechanism of Plasmonic Biosensors

Plasmonic optical biosensors are classified into two types of platforms; one which uses a thin metal-based film and another which uses nanostructure-based inorganic plasmon resonance [21]. The most common plasmonic biosensor used is the surface plasmon resonance (SPR) which is a metal-based film sensor, mostly made of gold, used to characterize biomolecular interactions [22,23,24]. The angle at which the SPR is formed as light is focused on the metal film and reflected onto a detector. The output collected is due to energized plasmonic electrons formed from the collective refractive index (RI) of the oscillations of the electrons in the conduction band and the oscillations of the electric field formed by the light. The angle at which the SPR is measured relies on the RI of the material attached to the metal film. Hence, any alteration in the angle of the incidence or the RI of the material affects the resonance measured while detecting the analytes [25,26]. Decreasing the size results in a blueshift which holds a high frequency, while increasing the surrounding dielectric constant results in a red shift which holds a low frequency [27,28,29]. As shown in Figure 2a, the targeted analytes in a sample bind to the biological receptors restrained on the film to form a different RI which is usually known and indicates the presence of the targeted analyte [30]. This SPR biosensor can detect viruses and then be reused after applying proper chemical treatment practices as shown in Figure 2b [31,32].

The efficiency of the plasmonic optical biosensor could be further enhanced when combined with surface-enhanced Raman scattering (SERS). SERS is used as a non-invasive, label-free diagnostic sensor that can attain abundant information from one measurement. Yet, due to the weak signal formed when detecting analytes at low concentrations, SERS is considered inappropriate [33]. Hence, by using plasmonic nanostructures for making the substrates for SERS, the signal becomes amplified by numerous orders of magnitude even though fabricating substrates for SERS is difficult [34]. This was proven in 2017 when Elsayed et al. fabricated a low-cost silicon substrate using silicon nanowires covered with silicon nanoparticles as substrates for SERS. Results showed detection levels reaching 10–11 M, which is a significant increase in the order of magnitude. The enhancement factor of silver NPs reaches 6–8 × 10^5^ but when deposited on the silicon nanowires, it reaches 10^11^ [35].

### 1.2. Determining the Efficiency of a Plasmonic Biosensor

To determine the efficiency of a biosensor, its limit of detection (LOD) and specificity(s) should be measured and attuned for its required application. The Clinical and Laboratory Standards Institute (CLSI) provides a guideline known as EP17 along with the protocols needed to determine the LOD [36] (p. 2).

#### 1.2.1. Limit of Blank

To measure the LOD with the highest precision, the limit of blank (LOB) needs to be identified. LOB is found by having a blank sample and measuring the highest value of the apparent concentration of analyte found not containing the targeted analyte and taking the average of several replicates of the sample. Although the sample used for finding the LOB would not contain the analyte being tested and should give no detected values, the blank sample can give an analytical signal which is consistent with low levels of the targeted analyte. Hence, to minimize the errors, LOB needs to be measured first. LOB can be found by measuring the replicates of the blank sample and then attaining the mean value of the result added to the standard deviation (SD) as shown in Equation (1) [37].
(1)$$\mathrm{LOB}={\mathrm{mean}}_{\mathrm{blank}}+1.645({\mathrm{SD}}_{\mathrm{blank}})$$
when having an assumption that there is a Gaussian distribution in the basic analytical signals taken from the blank samples, then the LOB would represent 95% of the values observed. The leftover 5% are a representation of the response which could be due to a very low concentration of analytes found in the sample. Such a false positive result is known as an α error. On the contrary, since a sample having analytes would exceed the LOB, a sample having extremely low concentrations of analytes could give a response below the LOB which is known as a β error [37].

#### 1.2.2. Limit of Detection

Characteristically, an assay cannot have a dynamic range starting from zero concentration to a certain limit. There must be “analytical noise”, which is the signal given due to the non-existence of the analyte [37]. LOD is the least number of analytes required to be detected by the biosensor efficiently, which is distinguished from the LOB [31,37]. Hence, the LOD is larger than the LOB. The LOD is estimated by measuring almost 20 replicates of a blank sample, defining the mean value and the SD, and then evaluating the LOD as the product of the mean +2 SD. The number of SDs can be increased to have a more conservative LOD. Therefore, this would be a successful and quick method since having a low analyte concentration will still give a signal which is higher than the LOB. The disadvantage of this method is that if there was a very low concentration of analyte present in the sample, it will be difficult to distinguish it from the signal produced from the blank sample. However, if the sample had a known concentration of the analyte, even if in small amounts, then the analytical response of this amount can be compared to that of the blank sample to conclude the analyte concentration required to recognize the presence of the analyte. Accordingly, the LOD would be more specific and meaningful and can be distinguished from the signal provided by the blank sample [37]. As determined in the EP17, the LOD can be defined based on the LOB and the replicated sample tests as shown in Equation (2) [36] (p. 17). The unit for LOD is in the range of micromolar (mM) to nanomolar (nM) which refers to the molar concentration of nanoparticles or the number of moles per liter in a solution [38].
LOD = LOB + 1.645 (SD_low concentration sample_)(2)

Once again, the Gaussian distribution is assumed providing 95% of the values to exceed the previously measured LOB with only 5% of the samples containing low concentrations giving values that are less than the LOB and inaccurately show no presence of the analyte. Therefore, manufacturers are required to determine the LOB and LOD using various reagents to provide the expected behavior of their analyzers. As provided by the EP17, when using 1.645 SD, a maximum of 5% of the values are expected to be less than the LOB. If the LOD being examined fits this criterion, then the LOD is verified. Otherwise, if the LOD shows values below the LOB, then the LOD is too little and should be re-estimated by testing samples having greater concentrations which will create a greater mean and hence a greater LOD [36,37].

#### 1.2.3. Specificity

The specificity (S) of a plasmonic biosensor is its ability to selectively detect a specific pathogen from a mixture having various materials. Measuring the specificity is calculated by dividing the variation in the peak of the wavelength reflected/transmitted (Δλ) by the variation in the concentration of the pathogen (Δc). The higher the result of the measured value, the greater the specificity as shown in Equation (3) [23].
(3)$$S=\frac{\Delta \lambda }{\Delta C}$$

Sensitivity or specificity could be further analyzed using hyperspectral imaging which consistently collects signals from more than 50 distinct nanoparticles in parallel. This will allow specific selectivity of the analytes with a narrow range of plasmon resonances while discarding the rest [39].

## 2. Applications of Plasmonic Optical Biosensors

### 2.1. The Use of Plasmonic Biosensors for Viral Detection

In 2019, an unanticipated disease known as the coronavirus (SARS-CoV-2) was discovered in Wuhan, China, causing a global pandemic. In less than two years, this novel virus affected millions and resulted in the death of more than four million people worldwide as per the WHO [40]. Furthermore, the coronavirus has significantly affected the world’s economy and resulted in a noticeable change in the social lifestyle of people to avoid getting infected with the fast-spreading, contagious virus. Hence, immediate detection methods of the virus with accurate, reliable, and instant results were needed to limit the virus from outspreading any further.

Infectious viruses have been detected over the years directly by targeting the virus itself or indirectly by targeting the secondary responses of the virus [41]. Targeting a virus directly involves targeting the entire virion, the antigens of the virus, or the single or double-stranded RNA or DNA of the virus. Indirect detection methods include serological testing for precise antibodies released due to a primary response to an antigen (IgM) or a secondary response due to previous exposure to the same type of antigen (IgG) [42].

Some of the most common methods in detecting infectious viruses include immunofluorescence assays, hemagglutination assays, viral plaque assay, viral flow cytometry, enzyme linked immunosorbent assay (ELISA) [43], chest computed tomography (CT) [44], and nucleic acid amplification test (NAAT) including polymerase chain reaction (PCR) and real-time quantitative PCR (RT-qPCR) [45,46,47,48]. Although those methods have shown successful outcomes, they have also displayed substantial limitations that deferred their usage in future viral detection as shown in Table 1.

Hence, to overcome those drawbacks, biosensors were pursued as viral diagnostic tools since they are highly-sensitive, affordable, robust, automated, and have a low fluid consumption with faster reaction time [53]. Viruses are detected using plasmonic biosensors that have planar, optofluidic, nanoparticle-based, quantum dot enhanced fluorescent local SPR, or nanowire-based structures as shown in Figure 3 [31]. Further description of each method will be discussed later.

#### 2.1.1. Planar Biosensors

Biosensors having SPR with planar structures are considered the most recognized since they are easily synthesized to detect one or more viruses. Some of the most common viruses detected by planar-structure biosensors include influenza A, influenza B, Dengue virus, Ebola virus, adenovirus, avian influenza virus, H1N1, parainfluenza virus 1–3 (PIV1, 2, 3), hepatitis B, respiratory syncytial virus (RSV), and one of the initial forms of coronavirus, the severe acute respiratory syndrome (SARS) virus [54,55,56,57,58,59,60]. All planar biosensors follow the same design of having a metal layer topped with a virus capturing layer to capture the targeted viruses.

To measure the SPR between the metal and the dielectric of the planar biosensor, the momentum of the incident photon and the conduction band electrons are measured. If both signals coincide and the analyte media has a high RI, light is coupled forming attenuated total reflection (ATR). This could be interpreted by using the dispersion relation by Raether (1988) shown [61]:(4)$$K_{spp}=\frac{2\pi }{\lambda_{0}}\sqrt{\frac{\varepsilon_{m}\varepsilon_{d}}{\varepsilon_{m}+\varepsilon_{d}}}=\frac{2\pi }{\lambda_{0}}n_{p}\sin\Theta_{i}$$

As shown in Equation (4), $K_{spp}$ represents the wave vector of the surface plasmon polariton (spp),$\;\Theta_{i}$ represents the angle of the incident light, $n_{p}$ represents the RI of the coupling prism, and the dielectric constants $\varepsilon_{m}$ and $\varepsilon_{d}$ represent the metal and the dielectric respectively.

In the previous equation, $\varepsilon_{m}$ is dependent on the incident light wavelength and $\varepsilon_{d}$ is dependent on the RI of the dielectric environment. As the RI of the media changes, $\varepsilon_{d}$ also changes and forms a variation in the wave vector, k. Hence, when $\varepsilon_{m}$ is equal and opposite to $\varepsilon_{d}$, k is optimized to its maximum which forms in a resonance. Accordingly, lots of configurations were made in the SPR planar structures including the Otto configuration, Kretschmann configuration, long-range SPR, plasmon-waveguide resonance, waveguide-coupled SPR, and the grating-induced SPR as shown in Figure 4.

The Otto configuration has a dielectric analyte separating the plasmonic metal layer from the prism as shown in Figure 4a. The Kretschmann configuration, which is the most frequently and commercially used, has the dielectric analyte on top of the plasmonic metal layer followed by the prism as shown in Figure 4b. The configuration shown in Figure 4c is known as the long-range SPR which has the prism topped with two dielectric analytes having the plasmonic material between them. Figure 4d represents plasmon-waveguide resonance where the plasmon dielectric waveguide is positioned between a metal layer and a less dense dielectric material. In Figure 4e, the waveguide-coupled SPR configuration has a thick plasmon dielectric waveguide sandwiched between two metal layers. The final configuration in 4f is the grating-induced SPR which is constructed to have a dielectric grating layer instead of the prism topped with a metal layer and then the dielectric analyte [62].

Studies were made in optimizing the LOD of planar biosensors. In 2019, Leuermann et al. conducted a study on planar waveguide-based interferometric biosensors since they are common to being highly sensitive, compacted in size, label-free, and have multiplexed processing. They also permit simple, fixed wavelength read-out which makes them suitable diagnostic monitoring devices with a low cost. The response of the biosensor to small alterations in the RI is dependent and the noise present in the read-put system determines the LOD. Hence, the group studied altering the order-of-magnitude in order to enhance the LOD to reach ~10^−8^ RIU using a silicon nitride biosensor working at telecom wavelength [63].

#### 2.1.2. Opto-Fluidic Nano-Plasmonic Biosensors

When nanophotonics and optics are integrated with optoelectronics, optofluidic structures are formed. Such structures allow fluidically engineering the plasmonic sensors to have highly sensitive optical bio-detection to accommodate for unrestrained tunable applications. When constructing optofluidic plasmonic biosensors, the plasmonic sensing properties of fluidic analytes for viral detection are improved by introducing nano-apertures. The nano-apertures contain capturing particles, such as aptamers or antibodies, to enhance the probability of the viral antigens binding with the antibodies. In 2010, Yanik et al. introduced nano-plasmonic biosensors which exploit the fluidic properties of various particles was made by having nano-apertures to capture and bind to diverse molecules. The biosensor had a metal dielectric layer with a 220 nm aperture for targeting and recognizing pathogens including small-enveloped RNA viruses (pseudo-typed Ebola) and large-enveloped DNA viruses (vaccinia virus) [64]. Consequently, optofluidic plasmonic biosensors started having abundant detection prospects to test for various pathogens.

In 2014, a more innovative work of using nanoapertures in optofluidic nanoplasmonic biosensors was made by introducing chips with programmable control systems that sense the microfluidic flow of analytes. These chips included a metal film with plasmonic pixels, each illuminated with two 11 nm apart light sources that emit light at 852 nm and 880 nm respectively. The diffraction patterns of the transmission signal of the nanoapertures changed whenever the RI of the analyte on the aperture changed upon binding. The higher the RI of the solution, the longer the wavelength shift of the plasmonic mode to be from 852 nm to 878 nm. Accordingly, the intensity of the diffraction signal of the nanoapertures changes as it decreases at 820 nm and increases at 880 nm upon LED illumination. This spectral shift from one plasmonic pixel increases the image contrast which allows the targeting of a broader range of different concentrations of analytes [65]. Subsequently, a more efficient and portable LSPR with a programmable system for microfluidic control of analyte flow was developed. In this work, nine different samples, each having its own RI and antigen-antibody system, were tested by the optofluidic array. This platform was then used to test the biomarker for liver cancer in situ and real-time and showed a detection of 45.25 ng/mL of its antigen and 25 ng/mL of its antibody [66].

In 2020, a discovery was made in detecting COVID-19 using an optofluidic structure with plasmonic sensing which included highly-specific serological testing as shown in Figure 5. The optofluidic device had pure gold nanospikes with a pre-calculated refractive index (RI). Upon exposing the microfluidic chip to different solutions, a shift in the peak of the LSPR wavelength was recorded. COVID-19 was then tested using the chip and antigen-antibody binding was declared when a shift in LSPR position was observed [67].

#### 2.1.3. Nanoparticle-Based Biosensors

Metallic nanoparticles (NPs), such as gold (Au) and silver (Ag), are used in plasmonic biosensors for detecting viruses since they are cost-effective, easily fabricated, and label-free. The SPR wavelength of gold NP is found at 520 nm while the LSPR wavelength of silver NP is found at 400 nm [68]. Nevertheless, the properties and wavelength of NPs can be modified upon altering their size or shape [69].

In 2013, the first Au NP-based plasmonic biosensor for viral detection was introduced. Immobilized antibodies were bound to the NPs to test unrefined whole blood for different types of HIV viruses at LOD of 98 ± 39 copies/mL for HIV subtype D. Since the wavelength peak of Au NP was known, the shift in wavelength resonance upon binding between the virus and the immobilized antibody on the surface of the biosensor was measured. HIV subtype A virus with a concentration of 6.5 × 105 copies/mL was detected at a wavelength shift of 9.3 nm [70]. In the same year, a study discussed using Ag Nanoisland (NI) based LSPR to detect adenovirus at LOD of 109 viruses/mL [71]. In 2014, another study used antigens and amines in the form of NPs on Au-coated glass substrate to detect four different types of Dengue virus at LOD of 10 antibody titers, 83–89% sensitivity, and 100% specificity [72]. In 2016, work was done to detect the respiratory syncytial virus (RSV) using anti-RSV polyclonal antibodies on Au, Ag, and copper (Cu) NPs. Among all types of NPs, Cu NP showed the best LOD with 2.4 plaque-forming units per sample [73].

In 2017, a hybrid nanostructure of Au NP and magnetic NP decorated graphene (MNP-GRP) was designed to target and bind to norovirus-like particles (NVLP). Binding of the virus at concentrations ranging from 0.01 pg/mL to 1 ng/mL with the NVLP antibody functionalized on the Au/MNP-GRP nanostructure was detected with LOD of 1.16 pg/mL [74]. In the same year, Au NPs in the form of a hetero-assembled sandwich were used to detect hepatitis B surface antigens (HBsAg) by functionalizing the antibody for HBsAg on the Au NP. The biosensor showed high specificity and the LOD was measured at 100 fg/mL [75]. Two years later, another work showed Au NP used to detect noroviral protein and human norovirus. Instead of using expensive antibodies, norovirus recognizing affinity peptides were used to detect the noroviral capsid proteins and showed LOD of 0.1 ng/mL for the noroviral protein and 9.9 copies/mL for the human norovirus [76].

Also, in 2019, LSPR for Au spike-like NPs was used for detecting AIV. This technique included detecting the virus with multi-functional DNA three-way-junction having the NPs, hemagglutinin binding aptamer, and thiol group. LOD in both surroundings, PBS buffer, and diluted chicken serum, was 1 pM [77]. Similarly, a biosensor made of thermally annealed Ag film deposited onto a silicon substrate with anti-nonstructural protein 1 (NS1) antibody was used to detect NS1 antigen of dengue virus from whole blood. The inlet of the biosensor had a filtering membrane made of polyethersulfone to separate the blood cells from the plasma. At 50 g/mL concentration, there was a 108 nm red shift in peak absorption wavelength, 9 nm sensitivity of LSPR, and LOD of 0.06 g/mL [78].

In 2020, two-dimensional Au nanostructures having octupolar geometry functionalized with rotavirus capsid (2B4) antibody were used for detecting and capturing low concentrations of rotavirus in water. For 105 PFU/mL of virus concentration, there was an LSPR wavelength shift of 46 nm and a LOD of 126 ± 3 PFU/mL [79]. In the same year, upon searching for a faster detection method for the more recent variant of the coronavirus, SARS-CoV-2, an experiment was done to detect the virus using a plasmonic biosensor uniting LSPR and plasmonic photothermal (PPT) effect. As shown in Figure 6, the biosensor was composed of functionalized two-dimensional Au NI to detect sequence-specific viral nucleic acid of SARS-CoV-2 and it consequently showed a LOD of 0.22 pM [80]. In the same year, work was reported on detecting SARS-CoV-2 using Thiol-modified antisense oligonucleotides capped gold plasmonic nanoparticles specifically for the N-gene of the virus. Consequently, it showed positive results within 10 min upon isolating the RNA samples [81].

#### 2.1.4. Quantum Dot Enhanced Fluorescent LSPR

Quantum dots (QDs) are luminescent semiconductor NPs with optical and electronic properties that vary depending on the size and type of QD [82]. Using QD in biosensors to help amplify the fluorescence signal for viral detection. QDs are composed of two main inorganic parts, a core, and a shell. They also have up to 100 nm of stokes shift which is the difference between the maximum absorption wavelength and the maximum emission wavelength. Unlike usual fluorescent dyes, QDs have a broad absorption that can be tuned and a significantly narrow emission spectrum. In 2012, the merging NPs with QDs began when Au NPs with QDs were used to detect AIV. This work was not tested using plasmonic techniques but it initiated the use of QDs for viral detection [83].

In 2016, LSPR for immunofluorescent nanobiosensor with introduced with Au NPS and CdSeTeS QDs to enhance the signal in detecting H1N1 influenza virus. For targeting and higher sensitivity, anti-neuraminidase antibody was conjugated with the Au NPs and anti-hemagglutinin antibody was conjugated with the QDs showing LOD of 0.03 pg/mL in deionized water and 0.4 pg/mL in human serum. It was also tested for clinically-isolated H3N2 virus and showed LOD of 10 PFU/mL [84]. In the same year, novel work was done in detecting NVLP using SPR-assisted CdSe-ZnS based QDs. The biosensor chip was held by a V-shaped trench. A wavelength of 390 nm was used to excite the SPR on the aluminum layer having a self-assembled monolayer (SAM) of phosphonic acid derivative to aid in immobilizing the proteins. A variation in the relative luminescent intensity indicated NVLP detection. Minimum LOD was noticed at 0.01 ng/mL which resembles 100 VLPs in the area of detection [85].

In 2021, a study was made involving the use of a cartridge SPR system controlled by a smartphone application that measures the dynamic absorption spectra. As shown in Figure 7, virus-specific antibodies were placed on a nanocup structure with a thin gold film coating. The antibodies were crosslinked with the SPR-based sensor chip to react with the SARS-CoV-2. Results showed high selectivity and low LOD of almost 4000 viruses [86].

#### 2.1.5. Nanowire-Based Biosensors

Nanowires are materials composed of one-dimensional nanostructures. Just like nanorods, they have a width ≥ 100 nm. However, the nanorods have a diameter of approximately 10–120 nm while the nanowires have lengths 1000 times more than their width. Both nanowires and nanorods are known for their exquisite magnetic properties. However, due to their high aspect ratios, nanowires have the privilege of possessing active thermal and electrical conductivity. The structure of nanowires is composed mainly of semiconductors, conductive polymers, and metal oxide components such as gold, copper, and silver [87]. The use of those noble metals in the nanowires helps in resonance scattering and absorption of light in the near-infrared region and the visible region upon plasmon oscillation excitation which makes them an exceptional choice for plasmon-related biosensors [88]. Accordingly, nanowires are integrated into biosensors to enhance their properties due to their capability of restraining electromagnetic fields immensely [89].

Furthermore, since nanowires have comparable sizes to various biological and chemical molecules, they were used as sensitive and selective detective means of biochemical reactions including surface binding reactions. Over the past few years, nanowires were used as biosensors to detect viruses where the nanowire biosensor is functionalized with antibodies with high specificity of binding to the targeted viruses which influenced the nanowire conductivity [90].

In 2010, biosensors made of silicon nanowires (SiNW) were used to rapidly detect Dengue serotype 2 (DEN-2) by identifying its reverse-transcription-polymerase chain (RT-PCR) product. This was done by covalently binding peptide nucleic acid (PNA) to the surface of the SiNW. RT-PCR amplification was done to obtain a complementary segment of DEN-2 which was introduced onto the functionalized PNA SiNW. The binding was verified via measuring the change in resistance of the SiNW prior to and after the binding process. Results confirmed that the SiNW-based biosensor can detect beneath a concentration of 10 fM of the amplicons within a timeframe of 30 min [91].

In 2011, bio-detection of one of the types of Influenza A, H1N1, was done using a multiplexed SiNW module for sequence determination. A PCR module having a microfluidic PCR chamber connected to an electrical controller was used for performing a multiplexed protocol which amplifies the nucleic acid targets and enriches specific fragments of H1N1 and any available influenza A strains with an amplification efficiency of $10^{4}$ to $10^{5}$. Then, the PCR amplicon was denatured and transferred to the SiNW module for detection after being introduced to a control SiNW to eliminate background interference. Once the target DNA having the virus was injected, a 10× variation in the magnitude of the current was observed. Results showed that the SiNW module had a sensitivity for H1N1 of 20–30 fg/μL and for influenza A of 10 μL. Having such a low sample consumption proved multiplexed modules to be highly sensitive and highly specific for the detection of both viruses [92].

In 2012, exhaled breath condensate (EBC) samples were used for the rapid detection of 29 viruses of influenza A/μL using a highly selective SiNW biosensor and an RT-qPCR for comparison purposes. The EBC collection device was designed to have four main parts: a cover for the collection device, a base for the collection device, a hydrophobic film, and a layer of ice to maintain the temperature of the hydrophobic film. Human subjects were asked to breathe into a straw connected to the device for 5 min where the exhaled breath rapidly condensed into water droplets on the hydrophobic film surface due to the low temperature. Then, 10 μL of DNA and RNA-free DI water was inserted and distributed onto the film using a pipette which was then transported to a sterile tube and stored at −70 °C. The negative control used in the device was the samples lacking the exhaled breath. Results showed that the SiNW biosensor was able to diagnose the virus with less time by two orders of magnitude in comparison to the RT-qPCR [93].

In 2017, a gate-controlled bio-memristor having SiNW in a honeycomb-shaped arrangement was used in the detection of Ebola VP40 matrix protein. The gate control was introduced to control and manipulate the voltage gap opening during measurement. This helped in providing the preliminary conditions for analytes with different charges to be detected. After 30 min of incubation, the biosensor had a LOD of 6.25 nM which outdid the ELISA method by six orders. The nanowire even included several advantages such as accuracy, rapidness, and ease of portability [94].

A study conducted in 2021 was used to detect the spike protein S1 of SARS-CoV-2 by using SiNWs as a substrate coated with plasmonic silver NPs via chemical processing. The NWs were etched for a longer time to enhance the length of the NWs from 0.55 μm to 0.55 μm. Results showed that the sensor can detect the S1 protein at a picomolar concentration of 9.3 × 10^−12^ M [95].

### 2.2. The Use of Plasmonic Biosensors for Environmental Evaluation

Plasmonic biosensors are great candidates when analyzing environmental contaminants. The amount of pollution increasing at a fast pace requires fast, highly specific, and cost-effective analytical tools that can be used for monitoring pollutants in our environment. Providentially, great initiatives have been made for controlling environmental pollution and several scientific researches were conducted and are still in progress to satisfy the concern of society regarding the environment and the pollution overtaking it [96,97,98,99]. Biosensors are great analytical techniques that can use a biological mechanism to detect analytes in the environment using chemical sensors for environmental evaluation [97,98,99]. When detecting environmental analytes, biosensors usually include whole microorganisms, DNA, enzymes, and antibodies as recognition receptors [100].

Environmental monitoring using biosensors rather than conventional analyzing tools is of great benefit since they are portable, miniatured, compact, and has high selectivity to different matrices using low input of sample preparation which can be used on an on-going basis for regular environmental analysis [101]. Hence, biosensors can be used as monitoring tools in the environment to assess the biological quality of ecological molecules including organic and inorganic pollutants [102]. In 2017, a MIR sensor was fabricated by using doped silicon structured as nanowires with 10 nm radius using a numerical analysis technique known as Finite-difference time-domain method (FDTD). The simulation done by the 2D FDTD showed a total-field scattered-field source with a 3 μm wavelength around the plasmonic resonance used for exciting the nanowire. As shown in Figure 8, when the refractive index of the environment changed, the sensitivity of the sensor was analyzed by calculating the extinction cross-section which includes the absorption as well as the scattering. When air is surrounding the nanowire, the resonance of extinction was almost 2.7 μm. Results have shown that as the refractive index of the surrounding increases by 0.25, the extinction cross-section increases by ~300 nm [103].

Furthermore, since one of the highly distinct properties of plasmonic biosensors is their optimum sensitivity and selectivity, plasmonic biosensors can withstand harsh environmental factors to be used for environmental monitoring and analysis with better precision. A paper published in 2019 and another in 2021 by Ghosh et al. amplified the sensitivity bar as the group synthesized a hybrid fiber-optic heavy-metal ion grating-based sensor that is highly flexible, compact, immune to external electromagnetic interferences, and unaffected by numerous chemicals to sense materials with high toxicity such as hormones, heavy metals, pesticides along with biochemical residues found in water which makes it greatly nominated to be applied in biosensors as well when targeting pathogens and biochemicals leading to diseases [104]. Furthermore, it was connected to an artificial neural network (ANN) for a sensor calibration graph measuring the concentration of the metal ion [105]. As shown in Figure 9, the optic sensor targeted selective sensitivity of heavy-metal-ions, mostly Pb (II), by joining in series a higher-order long period grating (LPG) having an attenuation band of LP_0,9_ along with a Fiber Bragg Grating (FBG) of 1524 nm. The FBG has high immunity to external alterations in the refractive index and hence was used as a reference point in terms of the static response of its attenuation band to measure the relative LPG shift accurately with an attenuation band (λ = λ_0,9_) wide peak and Bragg reflected wavelength (λ = λ_B_) small peak [106]. The LPG has a distinct light-guiding mechanism that allows surface-sensing without interfering with the structure of the fiber [107]. Layer-by-layer assembly of nanocomposite material along with 390 nm thick polyacrylic acid (PAA) was used to functionalize the LPG for optimal sensitivity. In addition, nitrogen-doped graphene oxide (NGO) with cross-linked chitosan polymer (CCS) and PAA were used for high sensitivity to the metal ions and 30-fold enhanced light-matter interaction via the reinforced evanescent field of the LP0,9. Analyzing the spectra was then done using an optical spectrum analyzer (OSA) and ANN system. Furthermore, information extracted from the LPG to test the multi-layer adsorbent coating were cross-referenced with Langmuir (LM), Holl-Krich (HK), Sips (SP), and Toth (TO) adsorption isotherm models. Moreover, the LOD of the sensor was shown to be 0.18 nM [106].

In 2020, Zhang et al. have also used SPR to detect heavy metals in contaminated water by fabricating a dual-channel optical fiber sensor. It was synthesized by initially sputtering silver with varying thickness on two fiber channels in order to have two different resonance dips with an RI sensitivity of 1334.56 nm/RIU and 1730 nm/RIU, respectively. Next, layer-by-layer functionalization of the thicker silver channel was performed using chitosan (CS) and polyacrylic acid (PAA) to detect the heavy metal ions present in the contaminated water. As for the thinner silver channel, it was considered as a reference to validate that the changes that occurred to the thicker wire were due to the heavy metals from the contaminated water or the surrounding refractive index. As the level of copper ions and other metal ions increased, a red-shift occurred in the resonance dip of the CS/PAA functionalized channel at low ion concentration. Results also showed that when the concentration was less than 80 μm, the binding of the copper ions to the CS/PAA layer was stronger, that it reached 0.249 nm/μM. This indicates that the fabricated sensor shows high sensitivity for copper ions at low concentrations [108].

One of the most recent highly sensitive optical sensors is plasmonic Mach-Zehnder interferometer (MZI) which is a label-free optical sensor with optimized sensitivity that could be easily fabricated, holds a long interaction-length, and is capable of phase measurement [109,110]. Several papers used plasmonic MZI sensors as lab-on-chip biosensors having a sensitivity attainment of 3695 nm/RIU at a wavelength of 730 nm and 57.6 μm device length [111,112,113]. Hence, MZI can be used as a fast and portable sensor that can work via multiplexed array sensing and the chip could be further enhanced by integrating it with a microfluidic channel [114].

Another paper published in 2020 by El Shamy et al. designed an on-chip MI optical gas sensor that could be used for detecting gases such as methane, carbon dioxide, and carbon monoxide using plasmonic MZI [115]. The sensitivity of the plasmonic sensor is enhanced due to the MI waveguide which develops a high-index dielectric layer above the metal after optimizing its thickness and refractive index. Two designs were proposed and tested. The first one had a highly sensitive sensor which enables it to perform exquisitely in wavelength and intensity interrogation systems. The second one functioned by minimizing the sensitivity to varying wavelengths causing the interrogation scheme to result in minimized cost and size. Results have shown the first design to have a sensitivity of 10,000 nm/RIU at a wavelength interrogation figure or merit (FOM_1_) of 133 RIU^−1^ and intensity interrogation of 239 RIU^−1^. As for the second design, the figure of merit was 363 RIU^−1^, a length of 250 μm, and a wavelength of 4.6 μm as shown in Figure 10 [115].

Two different types of on-chip gas sensor using metal-insulator (MI) plasmonic waveguide are shown in Figure 11 and Figure 12. They work in the mid-infrared range and utilize a Mach-Zehnder Interferometer (MZI). The MI waveguide utilizes a high index dielectric layer on top of the metal to enhance the sensitivity of the sensor. In Figure 11, the proposed structure consists of three layers: metal-sapphire-metal above a sapphire substrate, which forms MIM and MI waveguides that construct the MZI reference and sensing arms, respectively. The sapphire is chosen due to its low absorption in the wavelength range 1.1–6 µm, and the metal used is silver (Ag). The input plane wave, from the substrate, is coupled to the MIM and MI waveguide modes through the input slot of width w_1_, then each mode propagates with its propagation constant (β) distance L, and finally coupled out through the output slot w_2_ and interfere with each other. To improve the low sensitivity of the MI waveguide as well as low-index gas as the insulator material results in MI waveguide, a high index layer (HIL) is introduced above the metal of the MI waveguide, as shown in Figure 11a, forming MII waveguide. Where an optimum thickness and index for this layer is chosen to enhance the sensor performance. It has been found that for 250 nm thickness of this HIL, the highest sensitivity is achieved at an index of three for λ = 4 µm where silicon nitride Si_3_N_4_ has been used for the HIL layer with a refractive index of 2.4 around 4.5 µm wavelength [115].

With respect to the structure in Figure 12, the output transmission is low and a modification on the structure is performed to increase the transmission. Thus, a grating on the substrate-metal interface was used, as shown in Figure 12 to increase the input power coupling and hence the output power. The optimized grating dimensions are P_gr_ = 1216.2 nm, the grating period and h_gr_ = 475 nm, the grating thickness. Another optimization is performed on the input and output slots, where w_1_ = 1550 nm and w_2_ = 1600 nm. This enhanced the output power of the structure is shown in Figure 12, by a factor of 3.6. Then, further optimization is performed using FDTD simulations to maximize the figure of merit FOM and select the suitable operating wavelength. Note that, for the MIM to support single mode, the insulator layer thickness (D) must be lower than 1400 nm at 4.5 µm wavelength [115].

A great concern in continuous need for detection is the hormone-induced cancers that occur due to environmental estrogens interacting with the nuclear estrogen receptors resulting in a substantial hazard for humans and the health of most organisms [116].

In 2021, a group invented an optical biosensor having an enhanced light-matter interface with innovative surface-chemistry to detect endocrine disruptors in an ultra-sensitive manner. The biosensor is coated with highly tilted gold fiber Bragg-grating which works by exciting the high-density fine cladding spectral combs which intersect with the high absorption of the plasmonic biosensor for highly accurate and ultrasensitive analysis of the alterations that occur to the refractive index at the surface of the fiber. By using the estrogen receptors as representations, molecular dynamics were used to design a conjugate with estradiol and streptavidin to be able to have a surface-based affinity bioassay for detecting proteins instead of recognition of environmental estrogens only. The ultrasensitive plasmonic biosensor was able to detect the environmental estrogens at 1.5 × 10^−3^ ng/mL estradiol equivalent concentration level which is one order less than the maximum defined radioactive estradiol (17β-Estradiol) known as E_2_ in drinking water as per the Japanese government. Thus, making it one of the lowest detection limits for any estrogen receptor-based detecting technique reported till now. Adding this to all the other advantageous properties of plasmonic biosensors to detect the endocrine disruptors at ultra-high sensitivity is considered a revolution in environmental monitoring and analysis [117].

### 2.3. The Use of Plasmonic Biosensors for Food Analysis

Diseases and malnutrition occurring due to the quality of products made food safety of great priority to diminish the health risks [118]. Plasmonic biosensors are used in various applications as shown in Figure 13 [119,120,121,122,123].

Conventional methods used to ensure food safety like PCR, ELISA, high-performance liquid chromatography (HPLC), and liquid chromatography-mass spectrometry (LCMS) are accurate but costly, time-consuming, and laborious [124]. Hence, using automated optical biosensors is an optimum solution for analyzing food by using a highly sensitive and selective low-cost analytical tool [125]. SPR among different types of optical biosensors has undergone great development to enable the detection of various pathogens found in food. Plasmonic biosensors were further upgraded for better analysis via coupling of various methods (Raman, fluorescence, and luminescence) to have less LOD and higher sensitivity. Hence, as shown in Table 2, SPR biosensor is considered much more efficient than traditional methods in monitoring and analyzing food.

In 2019, Chaylan et al. studied the detection of Aflatoxin M1 (AFM1) which is found in milk, but is considered a mycotoxin. AFM1 is frequently found and considered extremely dangerous to humans. As per the European Union regulation, the maximum level of AFM1 accepted in milk is 152 pM for adults and 76 pM for infants. The research conducted was made on Si_3_N_4_ asymmetric MZI and has shown that when it was functionalized with antigen-binding fragments (Fab’), AFM1 could be detected in samples of milk. The asymmetric MZI sensors were able to detect a minimum concentration of AFM1 of 48 pM in samples having purified milk and concentrated milk. Due to the real-time detection, the binding that occurs between the ligand and the analyte allows further analysis of the kinetics of the reaction to determine the kinetic rate constants of the interaction [136] (p. 1).

## 3. Introducing Metamaterials to Plasmonic Biosensors

With all the exquisite properties of plasmonic biosensors, the sensitivity has been further improved by introducing metamaterials. Metamaterial-based biosensors provide different geometric structures, each having its own sensing properties, which expands and improves the use of conventional plasmonic biosensors [133]. In the 1960s, a Russian physic named Victor Veselago initiated a theoretical concept for materials with simultaneous negative permittivity and permeability where light propagates in an opposite direction to that of the flow of energy, giving an uncommon refraction of light. Materials that follow this concept are known as left-handed materials [134].

In 1999, a theory was made by Pendry et al. declaring that microstructures having extremely small nonmagnetic conducting sheets in comparison to the wavelength of radiation reveal a magnetic permeability that is highly effective and can be further modified to display changing magnetic permeability together with the imaginary component [135]. This substance was named in the same year by Rodger Walser as “metamaterials” which he defined as “macroscopic composites having a synthetic, three-dimensional, periodic cellular architecture designed to produce an optimized combination, not available in nature, of two or more responses to specific excitation” [136]. The following year, Smith et al. confirmed the use of left-handed metamaterial using a microwave regime by experimenting with interspaced nonmagnetic conductive split-ring resonators with continuous wires [137]. Since then, metamaterials have been used, manipulated, and geometrically enhanced to have tuned properties that can be used in different applications including sensors, biological imaging, and spectroscopy [138,139,140,141,142,143,144].

The use of metamaterials was further explored as biosensors as they were categorized based on their structure into three main groups; 2D metamaterial-based biosensors, 3D metamaterial-based biosensors, and meta-surface-based biosensors as shown in Figure 14. The breakthrough of metamaterials with their improved sensitivity allowed the successful detection of several viruses including HIV, Zika virus, avian influenza virus, CPMV, and PRD1 [145,146,147,148,149]. Even further, metamaterials became a tool for novelty in label-free point-of-care viral detection.

### 3.1. Two-Dimensional Metamaterial-Based Structures as Plasmonic Biosensors

A metamaterial surface based on quartz with a gold rectangular structure on it was designed as shown in Figure 15. The parameters of the inductance (L) and capacitance (C) were equivalent. Using a simple LC circuit, when the virus particles were introduced into the capacitor gap, the resonance frequency changed. Accordingly, the detection of various viruses was carried out by observing and detecting the alteration in the THz transmission spectra. Hence, bacteriophage viruses PRD1 and MS2 were detected at 60 nm and 30 nm, respectively. Sensitivity was measured to be 80 GHz/particle [144].

### 3.2. Three-Dimensional Metamaterial-Based Structures as Plasmonic Biosensors

Although 3D metamaterial-based biosensors are difficult to synthesize due to their complex 3D structure, several 3D metamaterials were designed and tested for their high sensitivity and were proven successful. In 2016, a 3D biosensor was made of silver-coated woodpile structure with 2600 nm/RIU sensitivity and with more than 3 × 104 degrees/RIU phase-sensitive response for analytes [146]. Another biosensor was designed in the same year by another group for a bulk 3D subwavelength structure. The design was made of grating coupled hyperbolic metamaterial which improves the angular sensitivity of a plasmonic-based biosensor for detecting cowpea virus. The device had a maximum sensitivity of 7000 deg/RIU [50,147].

### 3.3. Metasurface-Based Structures as Plasmonic Biosensors

In 2017, Ahmedivand et al. made a plasmonic THz metasurface having iron microstructures for a magnetic resonator and titanium microstructures for an electric resonator (torus) to design meta-atoms as asymmetric split-resonators for supporting ultra-strong and narrow magnetic toroidal moments in the THz spectrum as shown in Figure 16. Due to the sharpness of the toroidal moment, the sensitivity of the dip to a specific protein taken from the Zika virus envelope protein (ZIKV) which is attached to the plasmonic system was analyzed. Results have shown LOD of 24.2 pg/mL and a sensitivity of 6.47 pg/mL and the toroidal response line showed a very sharp, narrow, and deep shape [148].

Another THz metasurface-based biosensor tested in 2018 detected avian influenza viruses H5N2, H1N1, and H9N2. The metamaterial absorber used was working via Spoof Surface Plasmon Polariton (SSPP) Jerusalem cross apertures. Upon changing the alpha-beta parameters of each virus or changing the thickness of the analyte, a shift was detected in the absorption and resonant frequency. Accordingly, the subtypes of the avian influenza viruses were verified to be detected using this sensor [150,151].

## 4. Lab-on-a-Chip (LoC) for Plasmonic Biosensors

Molecular and serological testing is critical for diagnosing patients. Hence, the need for having faster and more reliable diagnostic tools for immediate disease analysis and prevention of the spreading of pathogens and diseases is always pursued and high in demand [152]. LoC is considered the best device used for Point-of-Care testing (POCT) as it is based on biosensors that are designed for their application and can be upgraded with the most recent advances using microfluidics [153,154,155].

In this section, plasmonic biosensors are focused on due to their various LoC applications. The main target of the plasmonic-based LoC devices is to use planar technology to make an integrated photonic circuit that has good sensing capabilities [156,157,158,159,160]. This technology has a few unique proficiencies which integrates many different sensors on the same chip to detect different pathogens. In addition, the utilization of planar technology supports the mass production need and provides cost-effective solutions. Hence, silicon-based photonics are known to be a major technology platform for such applications [161,162,163]. Hence, plasmonics have been recently introduced at a wide range for such applications due to their greater sensitivity and improved selectivity [113,164].

The integration of plasmonic materials such as gold, silver, and aluminum along with silicon photonics standard technology has been a challenge with two folds; the first one is technology-related where metals may cause contamination for the conventional standard silicon photonics technology, while the second is the ability to efficiently couple the light from dielectric waveguides with modal field ranges from few microns, in case of optical fiber, to few hundreds of nanometers, for silicon waveguide, to a plasmonic mode with tight surface confinement and modal field in the range of few nanometers only. The impact of the former challenge has been reduced over time by optimizing the fabrication technology and using materials that have both plasmonic effects and compatibility of the silicon technology such as TiN and ITO [165,166,167].

Good coupling can be achieved between a silicon waveguide and the plasmonic slot mode over a wide range of wavelengths using an orthogonal coupling scheme similar to Otto configuration [163]. In 2015, the ability of doped silicon to support plasmonic mode in the mid-infrared wavelength range was introduced for the first time [164]. The main mechanism was to control the doping level of the silicon to achieve a plasmonic wavelength within the mid-infrared range by doping for silicon. For instance, a doping ranging from 10^−19^ to 5 × 10^20^ cm^−3^ can achieve a plasmonic behavior starting from ~10 to ~3 microns, respectively. Other III-VI semiconductors have been recently utilized for such applications [167]. The III-V materials have superior advantages over the silicon such that the on-chip detector can be fabricated using a compatible material from the same material group. The MIR range also has the advantage of providing unique absorption peaks for sensing gases or liquids for biomedical applications. This added advantage can help increase the selectivity of the proposed sensing system. Hence, a plasmonic biosensor made only from a dielectric can be apprehended in the MIR using the planar fabrication technology and yet with high sensitivity and selectivity at the same time [103,115,168,169].

Optical confinement is preserved to a high extent when using doped silicon as an alternative for exciting plasmonic modes instead of using metals. In Figure 17, the optical field is confined near the surface in the case of doped silicon while being able to propagate for the entire length of the structure surpassing the propagation length of the silver-based structure. For the dopped silicon structure, it is shown that both interfaces have a tight confinement for the length of the entire structure on both the air–silicon interface and the silicon–silica interface [103]. While in the case of using silver, the optical field cannot preserve its confinement on the surface of the structure and starts to leak after a few micrometers on the side of the air–metal interface. On the other hand, SiO_2_–metal interface has negligible power coupled away from the SiO_2_–metal interface [167].

Another innovative solution was recently introduced through using all the silicon-integrated platforms while the silicon itself behaves like plasmonics [170]. More details regarding this technology will be given in the following section. For the latter challenge, various highly efficient optical coupling schemes between the dielectric waveguide and the plasmonics have been proposed in the last few years. For such a coupling mechanism, it is crucial to choose the proper plasmonic mode that represents high coupling efficiency with the counter dielectric one. For example, for direct coupling from optical fibers is always preferred to utilize long-range surface plasmon mode as it has a large modal area and good compatibility with standard single mode fiber in the IR range [171,172]. In 2018, Ayoub et al. designed a silicon-based plasmonic on-chip mid-infrared (MI) gas sensor that characterizes different gases, especially carbon-based. Highly doped silicon was used since it can behave as a plasmonic media which combines two interfering surface waves to the output waveguide. The upper part of the surface acts as the sensing arm for the MZI while the bottom part of the surface acts as the reference arm for the sensor as shown in Figure 18. Results showed high sensitivity reaching 16,000 nm/RIU at a wavelength of approximately 5100 nm. Also, the sensor was built at complementary metal-oxide-semiconductor (CMOS) compatible materials which are the standard industrial-scale process responsible for making integrated circuits, and hence, caused the designing process of the sensor to have a small footprint [113].

Research conducted by Schwarz et al. proposed monolithically integrated mid-infrared lab-on-a-chip plasmonic structures for chemicals detection (e.g., H_2_O/C_2_H_5_OH) with different concentrations. In their work, they demonstrated a mid-infrared on-chip sensor based on absorption spectroscopy. The monolithic device comprises a laser, an SPP waveguide as well as a detector integrated on a single chip. A bi-functional quantum cascade laser/detector (QCLD) has been used for the laser and the detector. The active region has been used either as a laser or a detector depending on the applied bias. The laser light is coupled through an SPP waveguide, acting as an interaction zone, to the detector. Due to the evanescent nature of SPPs, 96% of the mode stays outside and interacts with the chemicals under test (analyte). The waveguide is fabricated with a thin dielectric stripe (200 nm) on top of an un-patterned gold surface. The fabricated devices with 50 and 100 µm waveguide lengths show coupling efficiencies (from the laser to the detector) of 30–50%. At room temperature, the laser has a peak output power of 200 mW and the on-chip detector has an internal quantum efficiency of 33%. Owing to direct coupling from the waveguide and the optimized quantum design, the on-chip detector provides a better performance than discrete QCDs. The laser light is emitted mainly to free space and partly into the substrate. By adding a gold layer in the direct vicinity of the laser facet, a surface plasmon can be excited. As this SPP is weakly confined (50 µm), it cannot be coupled efficiently into a ridge detector. One has to squeeze the SPP by an order of magnitude to provide a high mode overlap required for efficient end-fire coupling. This can be achieved by applying a thin SiN_x_ layer on top of the metal that enables the excitation of an SPP, which is strongly bound to the interface. In this case, the SPP can be directly coupled into the ridge detector with increased efficiency. The optimal thickness of the SiN_x_ layer in terms of coupling efficiency lies in the same range as required for proper waveguiding (200 nm SiN_x_). As for the coupling efficiency, it describes the ratio between the power through the detector facet and the power emitted from the laser facet. Without any waveguide, the coupling efficiency drops cubically with increasing distance. With a gold stripe between the laser and the detector, the coupling is improved, but still shows a significant drop at larger waveguide lengths. For the DLSPP waveguide, the coupling efficiency also initially drops because of additional free space coupling, but then approaches a constant slope determined by the waveguide loss. There is a significant dependency on the gap between the ridge facets and the SPP waveguide. For a 50-µm DLSPP waveguide and a gap between the gold surface and the ridge facets of 0.5 and 2.5 µm, the coupling efficiency has been calculated to be 47% and 35%, respectively. All curves in the results provided showed an oscillation because of the presence of longitudinal modes within the SPP waveguide. The on-chip detector signal and the laser power versus the laser current density for the device with a distance of 50 mm was measured and the inset showed the time-resolved detector signal of a single pulse at the maximum laser output power [168].

For gas sensing applications, Zaki et al. proposed an integrated optical sensor using hybrid plasmonics for lab on chip applications. The structure comprised a plasmonic resonator based on a rectangle cavity coupled to an MIM waveguide. Numerical simulation results show that high-sensitivity sensors can be created by the proposed structure. This is due to the strong overlap of the hybrid plasmonic mode with the sensed gas in the cavity. Through optimization of the cavity dimensions and other design parameters, sensitivity of 1500 nm/RIU reached a wavelength of almost 1.55 μm. The proposed plasmonic structure offers potential applications in integrated on-chip sensors as shown in Figure 19a,b. A thin silicon oxide layer of thickness *g* is sandwiched between two silver regions to create an MIM waveguide. A rectangular cavity of length *l* and thickness *t* is located inside the silver region on one side of the MIM waveguide. The gap between the cavity and the waveguide is a silicon layer of thickness *s*. Along the coupling distance L_c_, the electromagnetic wave is guided in the hybrid plasmonic waveguide composed of silicon, silicon dioxide, and silver. Contrary to the MIM waveguide where the electromagnetic field is strictly confined in the oxide, in the hybrid plasmonic section, part of the field can penetrate the silicon layer. The related spectra and sensing characteristics are shown in Figure 19c–e where the usage of the proposed coupling technique allows the shrinkage of the cavity size down to submicron dimensions without sacrificing its sensitivity [160].

## 5. Future Perspective

Pathogen detection has always been of great interest, especially when looking for less invasive and more rapid, sensitive, and accurate techniques. Devices providing those specifications to be used for point-of-care pathogen recognition are in great need, especially with the increasing number in population and the continuous emerging of pandemics that require immediate detection, especially if they are highly contagious. Biosensors based on nanomaterials were recently introduced and their integration with plasmonic detection resulted in highly-sensitive, highly accurate, and reliably fast results [80].

Designing a plasmonic biosensor based on machine-learning algorithms starts by analyzing several detection setups to sense various DNA oligomers, which are short single-strands used to detect specific sequences. Then, machine-based algorithms are tailored to evaluate the performance of the metamaterial-based plasmonic biosensor. Such an integration would intensely increase the sensitivity and real-time SPR responses in the detection process [169]. Another approach for more efficient plasmonic biosensors was conducted by using artificial neural networks to generate a connection between plasmonic geometric parameters with the resonance spectra. As a result, the spectra of a tremendous amount of varying nanostructured biosensors can be projected [173].

The application of 3D topological insulators has also been of great interest in biosensors. Those materials have exotic properties arising from their quantum nature which makes them behave internally as insulators and externally as conductors, allowing electrons to move only on their surface [174]. Plasmonic biosensing aided by quantum-based light properties would intensify the sensitivity of the sensor as tested by a group in 2018. In this research, the plasmonic biosensor was probed by bright entangled twin beams to measure local variations in the RI. Results have shown that the sensitivity has improved by 56% in comparison to classical configuration and by 24% in comparison to optical single-beam configuration [175].

Another evolving field in the recognition of pathogens is surface plasmon resonance imaging (SPRi) which was tested for detecting apple stem pitting virus by detecting the binding of the aptamer and the coat protein using SPRi [176]. Even more, this method was used to develop an even more advanced imaging technique known as plasmonic nanoaperture label-free imaging (PANORAMA) which detects dielectric nanoparticles based on unscattered light. This procedure could determine the size, number, and availability of nanoparticles past 25 nm and measure their distance from the plasmonic surface within a timeframe of a few milliseconds [177]. The detection limit of the metamaterial-based plasmonic biosensors sensitivity has reached femtomolar but has affected its specificity. Accordingly, the use of binding molecules made of aptamers and peptides was used to significantly enhance the specificity but showed to be applicable to some biosensors, especially when tested using clinical samples. Another matter of concern in the mass production of plasmonic biosensors based on metamaterials is the cost of synthesis opposed to the sensitivity measurement [178].

Another team of researchers designed a plasmonic biosensor based on metamaterials to detect pathogens from a gaseous environment using optical and thermal means. The sensor is made of gold nanoislands placed on a glass substrate with artificially produced DNA receptors grafted onto them to reliably identify the unique RNA of the virus. LSPR is used for detection as the optical sensor is located at the back of the sensor. Results have shown high accuracy and reliability [179]. In another recent work, the synthesis of plasmonic biosensors using nanomaterials, not metamaterials, was implemented for testing the coronavirus from liquids and gases. The researchers developed a sensing device connected to a genetic algorithm intelligent program that automatically designs and optimizes the device for ultra-high sensitivity of 1.66%/nm, a widespread recognition range, and can be used in liquid or gaseous environments. Furthermore, the exceptional infrared fingerprint detection features enable the sensor to detect mutated viruses, hence providing a label-free, multifunctional, ultrasensitive, and rapid diagnostic tool for pathogens [180].

Finally, a highly needed future development for plasmonic biosensors is a reduction in size and greater simplicity for better analysis in different fields, whether in vitro or in vivo. This would widen the scope of the application of plasmonic biosensors while having a long-lasting impact, since they are fast and low in cost, which would guarantee better monitoring and analysis as a biosensor.

## 6. Conclusions

Treatment against pathogens and infectious diseases requires cost-effective, highly specific analytic techniques to enable the identification of such microorganisms for a better state of health. Immediate and innovative diagnostic techniques such as optical biosensors using LSPR were made for the detection of diseases, environmental pollutants, and monitoring and analyzing biomolecules in food, and showed high selectivity and efficiency. However, the pursuit for more reliable, sensitive, and rapid detection is constantly needed due to novel and mutated pathogens and viruses that may appear at any time. Nanoparticle-based plasmonic biosensors have shown exceptional properties when used in pathogen detection due to their high sensitivity and low LOD, which enable the detection of various diseases due to the broad range of antibody binding. They are also highly needed in POCT as they are non-invasive, rapid, and accurate when used as biosensors. By introducing metamaterials to plasmonic biosensors, the sensitivity increases even further, allowing the biosensor to be robust and reproducible. Several researchers have been recently working on upgrading the biosensors to be developed as a lab-on-a-chip diagnostic tool, which would make it omnipresent. Furthermore, other researchers are working on using it to detect airborne diseases in the environment. Once research overcomes these limitations, metamaterial-based plasmonic biosensors would enable highly accurate rapid detection of pathogens that can analyze food products for better human well-being and prepare humanity against any pandemic in the future, regardless of their means of transmission, physical or airborne.

## Figures and Tables

**Figure 1 biology-11-00621-f001:**
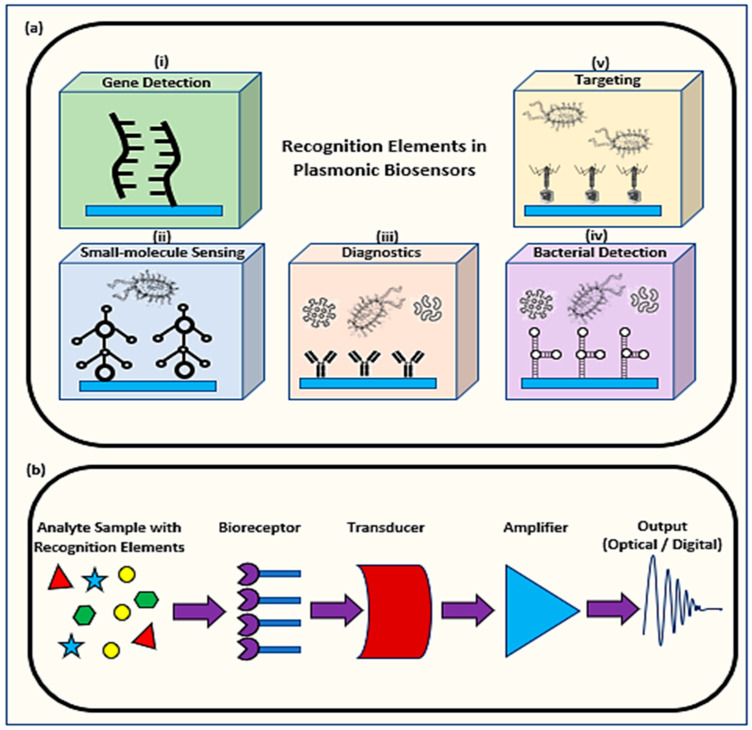
A schematic diagram of (**a**) recognition elements in plasmonic biosensors in (i) gene detection between electrochemical DNA and target DNA, (ii) small-molecule sensing for analyte-antibody conjugation, (iii) diagnostics between antibodies and analytes, (iv) bacterial detection between receptor biomolecule and bacteria, (v) targeting specific biomolecules and (**b**) the parts forming biosensors including a bioreceptor to capture the analyte connected to a transducer to convert the sample to be amplified and digitally presented.

**Figure 2 biology-11-00621-f002:**
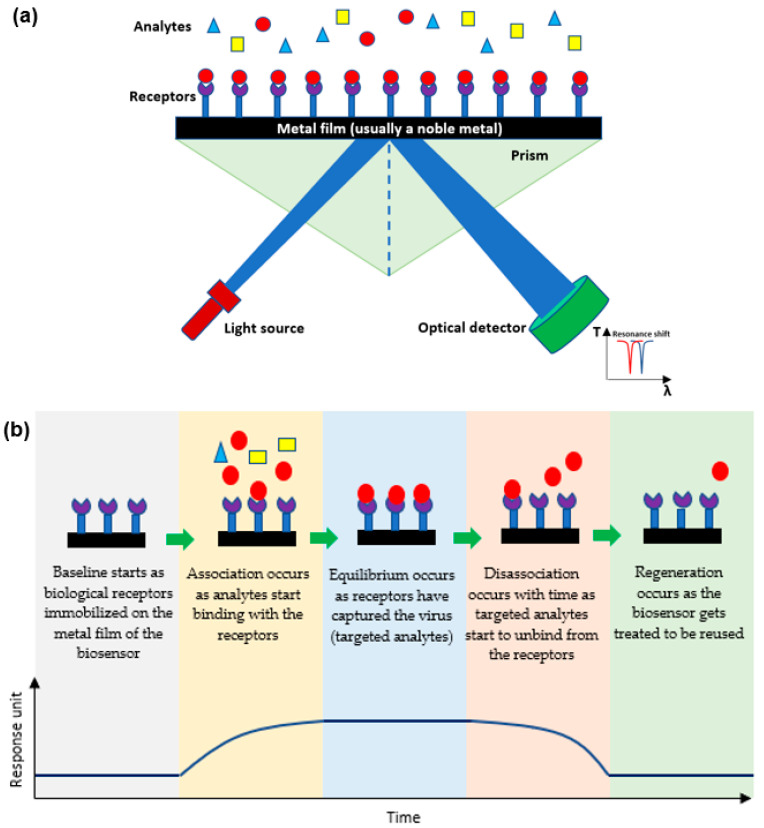
(**a**) A schematic diagram of the mechanism of plasmonic optical biosensors, and (**b**) Stages of SPR sensor from detecting analytes to detaching to be reused.

**Figure 3 biology-11-00621-f003:**
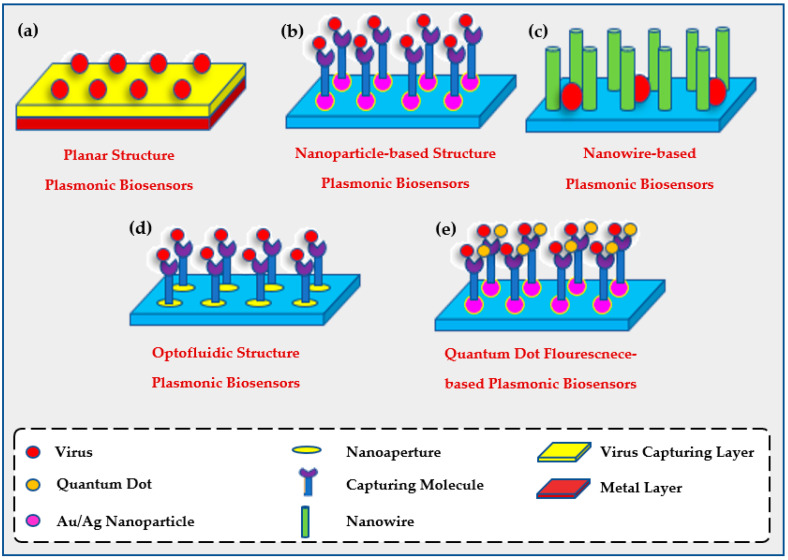
Common structures of plasmonic biosensors including (**a**) planar structure plasmonic biosensor where the base is a metal layer covered by a virus capturing layer to detect the viruses, (**b**) nanoparticle-based structure plasmonic biosensors bind to metal then a capturing molecule to capture viruses, (**c**) nanowire-based plasmonic biosensors with nanowires to entrap the targeted viruses, (**d**) optofluidic structure plasmonic biosensors with a virus capturing layer bind to a capturing molecule for higher detection of viruses, and (**e**) quantum dot fluorescence-based plasmonic biosensors with the capturing molecule bind to quantum dots which show visible changes among binding with viruses.

**Figure 4 biology-11-00621-f004:**
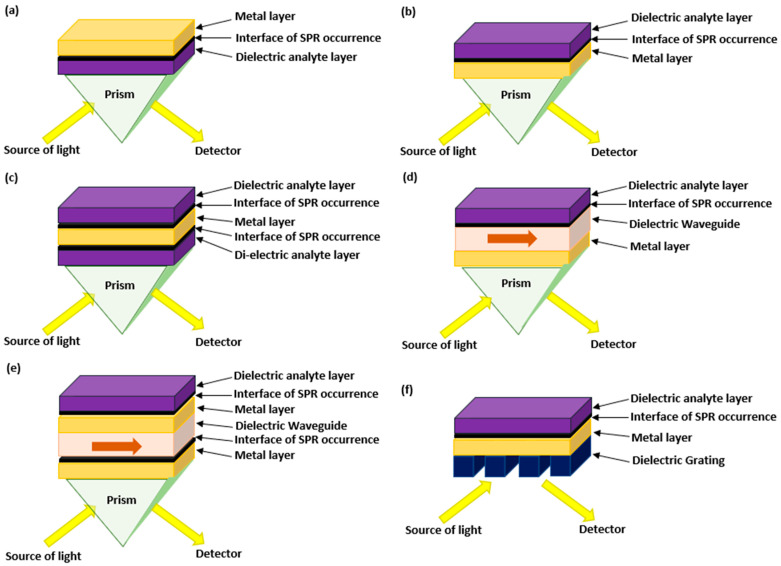
Common configurations in the SPR planar structures including (**a**) Otto configuration, (**b**) Kretschmann configuration, (**c**) long-range SPR, (**d**) plasmon-waveguide resonance, (**e**) waveguide-coupled SPR configuration, and (**f**) grating-induced SPR.

**Figure 5 biology-11-00621-f005:**
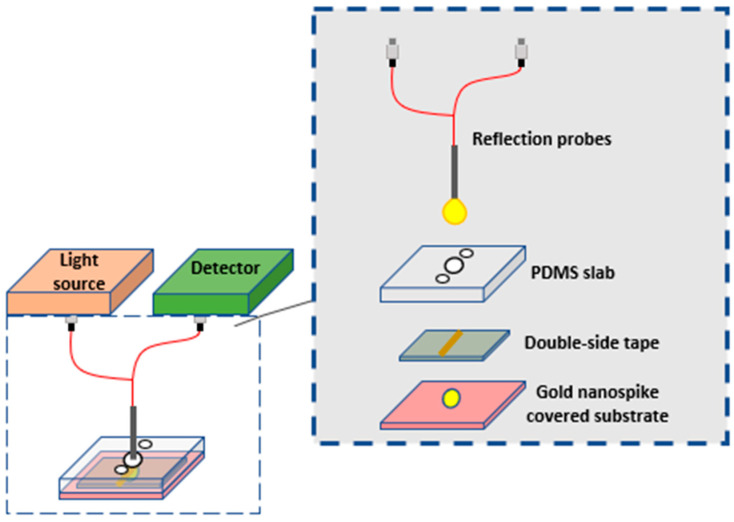
Schematic diagram of the composition of the integrated optofluidic device where the light source gives light to the reflective fiber optic probe to excite the gold nanospikes implanted in the opto-microfluidic chip. The probe also gathers the reflected light back to the detector.

**Figure 6 biology-11-00621-f006:**
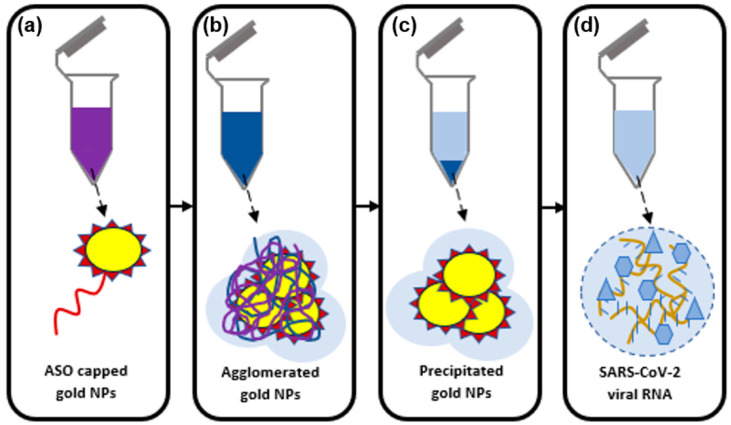
Schematic diagram of (**a**) ASO-capped AuNPs targeted for N-gene of SARS-CoV-2, (**b**) agglomerated AuNPs in the presence of the virus RNA, (**c**) selective positive detection of SARS-CoV-2 with the visual naked-eye, and (**d**) extracted SARS-CoV-2 viral RNA.

**Figure 7 biology-11-00621-f007:**
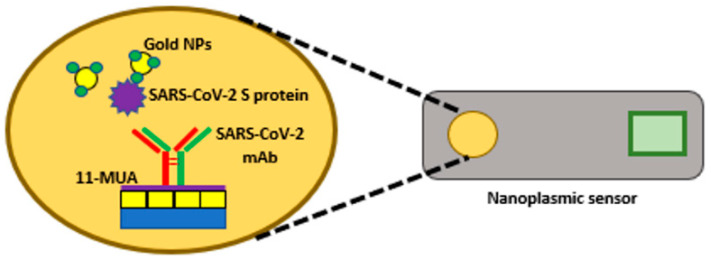
Schematic diagram of nano-plasmonic sensor chip cartridge used to detect SARS-CoV-2 using a sensor chip cartridge.

**Figure 8 biology-11-00621-f008:**
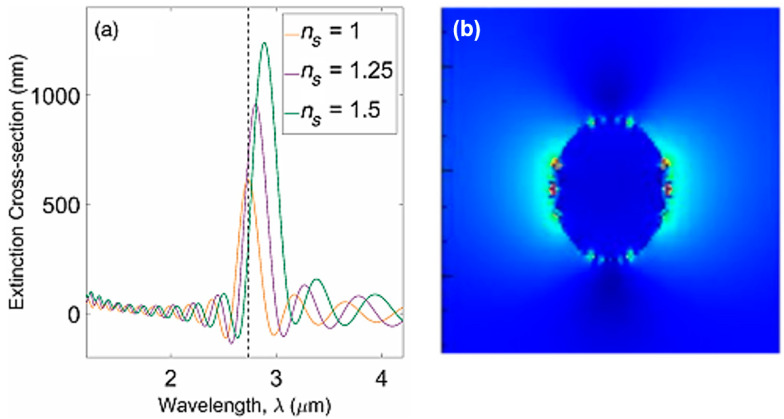
(**a**) Silicon-doped NW sensor with plasmonic resonance at 3 μm, having surrounded refractive index, n = 1, 1.25, and 1.5 in the surrounding. (**b**) Mode profile at 3 μm at the cross-section of the nanowire. The near field intensity (color) in (**b**) shows the maximum field in the center of the nanowire being shown at the cross-section. Reprinted with permission from Ref. [103].

**Figure 9 biology-11-00621-f009:**
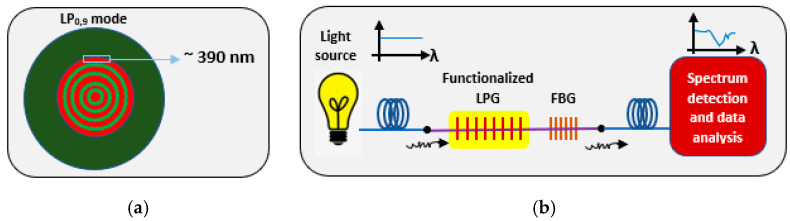
Schematic diagrams of (**a**) LP_0,9_ mode and of (**b**) fiber optic hybrid grating setup via butt-coupling an FBG.

**Figure 10 biology-11-00621-f010:**
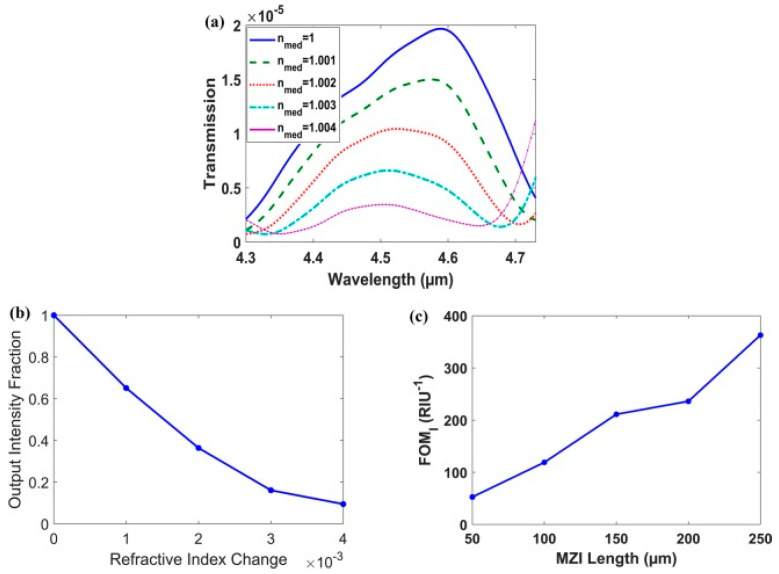
Vertical plasmonic MZI with D = 1000 nm, T = 2000 nm, h_HIL_ = 320 nm, w_1_ = 2100 nm, w_2_ = 1800 nm, P_gr_ = 918.4 nm and h_gr_ = 450 nm. (**a**) Transmission spectrum at different medium indices at L = 250 µm. (**b**) Output Intensity Fraction around λ = 4.6 µm versus air medium refractive index change (Δn_med_) at L = 250 µm. (**c**) Intensity interrogation FOM_1_ versus MZI length around λ = 4.6 µm. Reprinted with permission from Ref. [115].

**Figure 11 biology-11-00621-f011:**
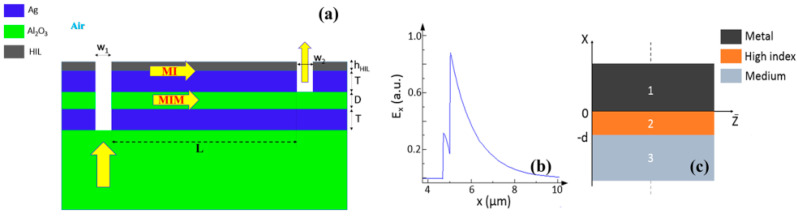
Vertical plasmonic gas sensor MZI with high index layer (HIL) (**a**) structure such that h(HIL) is the high index layer, T is the metal thickness, D is the insulator thickness, w_1_ is the width of the input slot, and w_2_ is the width of the output slot, (**b**) MII waveguide mode major component E_x_ with silver and HIL with index 2.4 and thickness 320 nm at λ = 4.5 µm with metal/HIL interface at x = 4.7 µm, (**c**) MII waveguide [115].

**Figure 12 biology-11-00621-f012:**
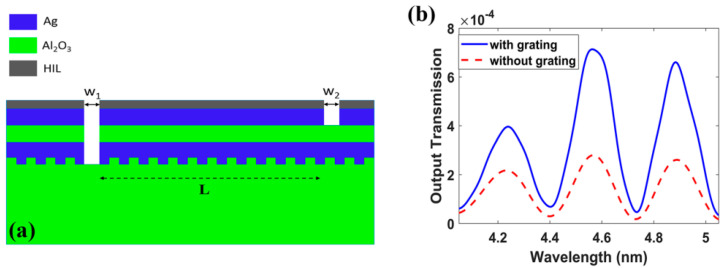
Vertical plasmonic gas sensor MZI with high index layer and grating (**a**) Structure, (**b**) Normalized Output transmission versus wavelength with grating P_gr_ = 1216.2 nm, h_gr_ = 475 nm and without grating with D = 900 nm, T = 1500 nm, h_HIL_ = 240 nm, w_1_ = 1550 nm, w_2_ = 1600 nm and L = 250 µm [115].

**Figure 13 biology-11-00621-f013:**
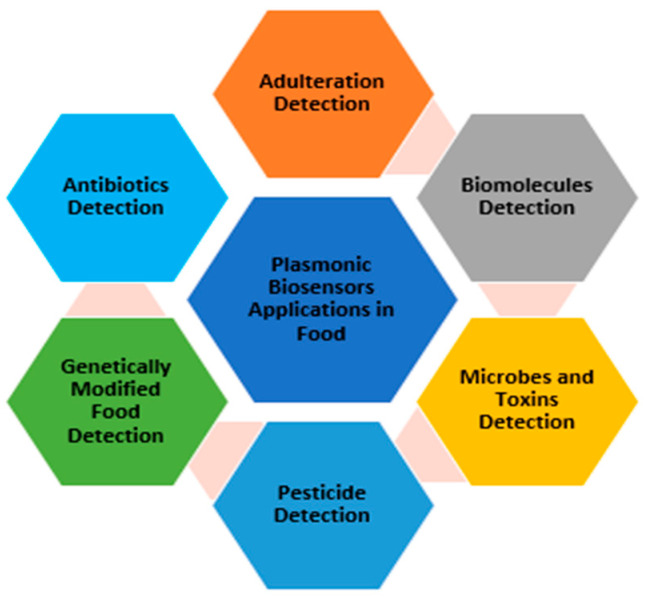
Schematic diagram of applications of plasmonic biosensors in monitoring food.

**Figure 14 biology-11-00621-f014:**
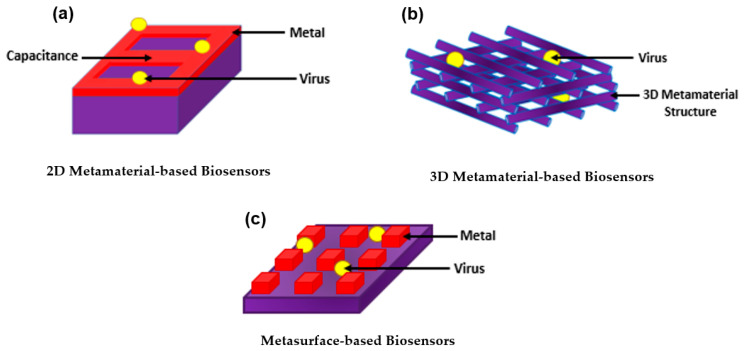
Common categories of metamaterials as (**a**) 2D metamaterial-based biosensors; (**b**) 3D metamaterial-based biosensors; and (**c**) metasurface-based biosensors.

**Figure 15 biology-11-00621-f015:**
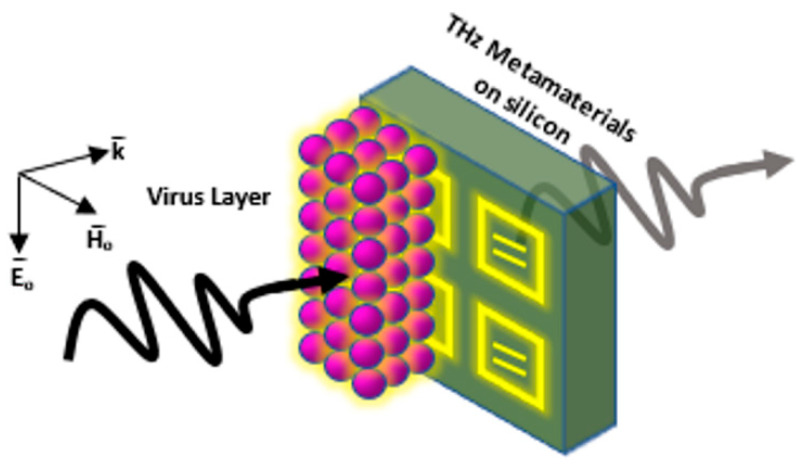
Schematic diagram of nano-gap gold metamaterial based on LC circuit based on dielectric constant virus detection and measurement via THz metamaterials.

**Figure 16 biology-11-00621-f016:**
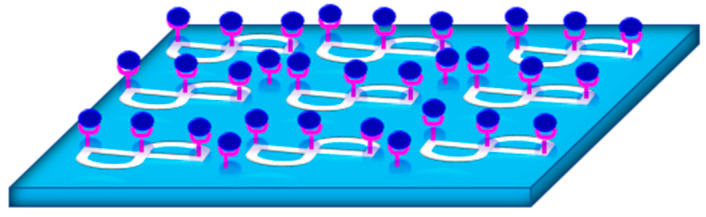
Schematic diagram of ZIKV envelope protein binding with their matching antibodies on toroidal THz plasmonic metasurface.

**Figure 17 biology-11-00621-f017:**
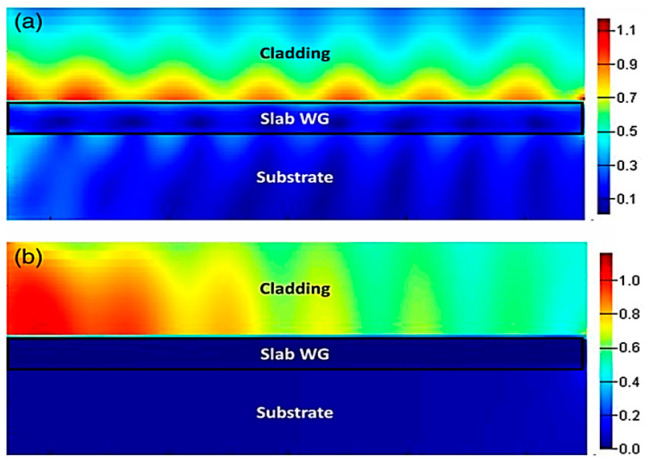
Comparison between the confinement capability for a test length of 400 nm for a (**a**) dopped-silicon slab and (**b**) silver slab on a SiO_2_. The total propagation distance is 10 μm. Reprinted with permission from Ref. [103].

**Figure 18 biology-11-00621-f018:**
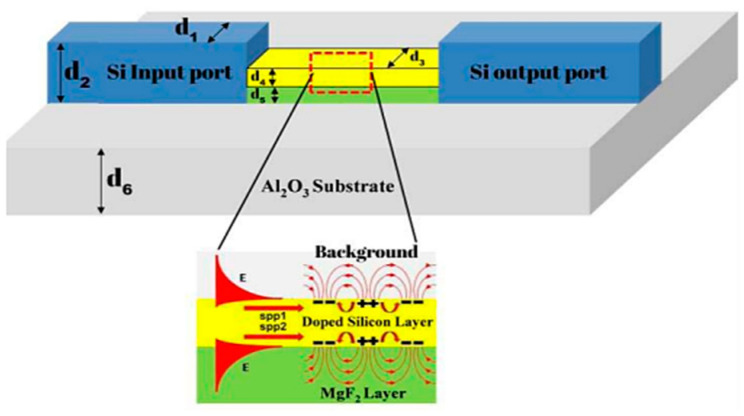
Schematic diagram of the 3D-design of the plasmonic MZI, d_1_ = d_3_ = 1.4 μm, d_2_ = 0.8 μm, and d_6_ = 1 μm. The insertion shows the electric field distribution on the insulator-metal-insulator interface. It also demonstrates the electric field exponential decay across the doped-silicon depth along with the propagating path as shown by the exponential curve. As for the background, it refers to the sensing arm. Reprinted with permission from Ref. [113].

**Figure 19 biology-11-00621-f019:**
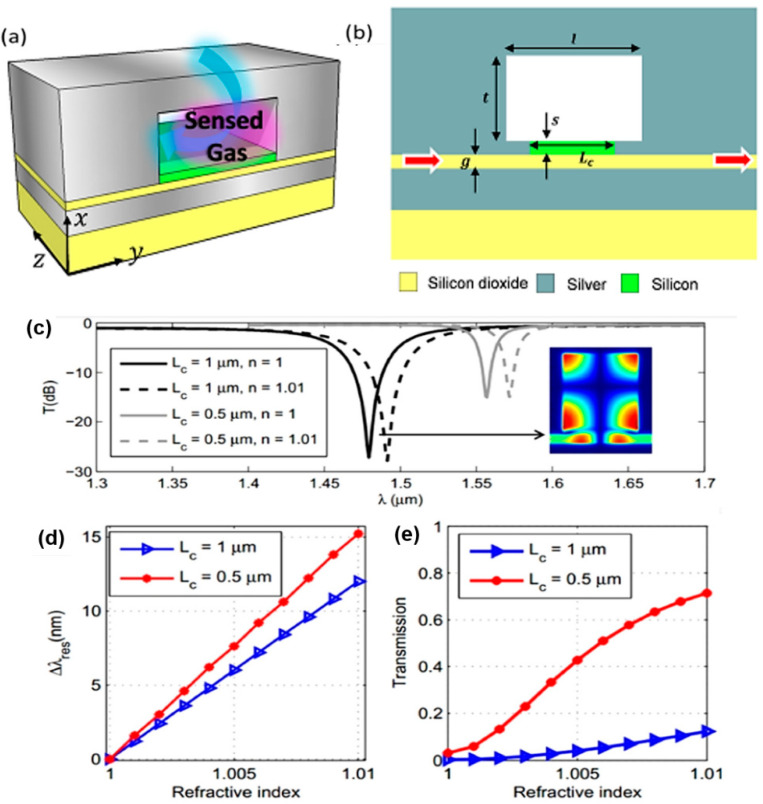
Schematic diagram of the plasmonic sensor with geometrical parameters symbols: (**a**) 3D illustration of the resonator in sensing application, (**b**) 2D sensor schematics, sensing characteristics of the 1 μm × 1 μm cavity of (**c**) Transmission spectra for different refractive indices of the cavity filling material and different coupling lengths (L_c_), inset showing the magnetic field profile of the resonant mode for L_c_ = 1 μm. (**d**) Resonance wavelength versus the refractive index. (**e**) Sensor’s transmission versus refractive index at the resonant wavelengths. © IOP Publishing. Reproduced with permission. All rights reserved. [160].

**Table 1 biology-11-00621-t001:** Advantages and limitations of common viral detecting methods.

Current Methods for Virus Detection	Advantages	Limitations	Refs.
**Immunofluorescence** **Assays**	Numerous, simultaneous samples can be analyzed and stored for some time.	Fluorescent molecules bound to primary antibody is limited. Low sensitivity may result in false negatives.	[49]
**Hemagglutination Assays**	Low-cost instruments. Results within hours. Has standardization as it is recognized in labs worldwide.	Little specificity. Requires trained personnel. Analysis needed by qualified individuals. Difficult inter-laboratory comparison of results due to the several controlled variables.	[50]
**Viral Plaque** **Assay**	Available in most labs. Rapid results.	Absence of standardization. Involves costly repeat runs for accurate results.	[51]
**Viral Flow** **Cytometry**	Rapid results. Numerous cells analyzed instantly.	Requires highly trained personnel. Requires ongoing maintenance by service engineers.	[52]
**ELISA**	Accurate/fast results. Very sensitive process. Easily automated.	Expensive preparation method. Requires trained personnel.	[43]
**CT**	Combined assessment. Short acquisition time.	Expensive preparation method. Requires trained personnel. Exposure to gamma rays.	[44]
**NAAT**	Very sensitive process. Accurate and reliable	Requires trained personnel. Expensive detection kits. Time-consuming (2–3 days). False-positive cases.	[46,47,48]

**Table 2 biology-11-00621-t002:** SPR in comparison to other techniques for monitoring food.

Component	Other Methods	SPR	Refs.
**Heavy metals**	Atomic Absorption Spectroscopy Destructive technique and single sample analyzed Inductively Coupled Plasma Mass Spectrometry Costly and destructive technique X-ray Fluorescence Spectrometry Radioactive technique	Low-cost Quick measurement Highly sensitive Non-destructive	[126,127]
**Food Allergens**	ELISA Varying as per the type of kit with LOD is 2.5 mg/L	Highly sensitive LOD of 57.8 ng mL^−1^	[126,127]
**Citrinin** **(Mycotoxin)**	HPLC and LC-MS Time-consuming and costly	Highly effective and selective Simple, quick, and highly sensitive	[128,129]
**Pesticides**	LC-MS/MS Complicated Requires sample pre-treatment prior to analysis	Highly precise Less response-time Low-cost and low LOD	[130,131]
**β-Lactoglobulin**	ELISA and LC-MS Inconsistency and costly	Speedy and detects in real time Resistance to environmental factors	[132,133]
**Tetrodotoxin** **(Fish toxin)**	LC-MS/MS, ELISA, and HPLC Costly and time-consuming	Speedy Low LOD	[134,135]

## Data Availability

Not applicable.

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
