# Peer review of "Plasmonic Biosensors: Review"

_biology, 2022, doi:10.3390/biology11050621_

Round 1
Reviewer 1 Report
The review by Hamza et al., summarizes the designs and applications of plasmonic biosensors. The manuscript introduces related publications in detail. Overall, the review is well-organized and useful to the community. Several edits are recommended before potential publication.
1) Page 2, line 8, more references are recommended for the two categories of biosensor materials.
2) Figure 1, the author should explain the meaning of all the patterns (like the bacteria-shaped pattern) drawn in the figure. The details should be included in the figure description.
3) Figure 2 may not be necessary to summarize the advantages of plasmonic and metamaterial-based sensors. The author may summarize the benefits in a table with references, or in the main text.
4) Page 3, line 80, please add references for surface plasmon resonance (SPR).
5) Page 4, line 114, in the section for efficiency determination. The author is encouraged to introduce how limit of detection (LOD) and specificity (s) are measured experimentally.
6) Page 5, line 152, please add reference for all the biological assays mentioned here for detecting infectious viruses.
7) Figure 4, the author should write the figure description in a detailed manner, with brief introduction of all mentioned architectures.
8) Page 9, line 301, when referring to a previous publication, in addition to the year of publication, the author could also mention the name of the first author of that publication.
9) Figure 7, please explain the workflow in figure description in a more detailed manner.
10) Figure 9, the author may want to rearrange the size of fig 9(b) and illustrate how the scattering pattern is related to fig9(a).
11) Figure 12, the author should explain the meaning of the letters (h, t, d, w) used in the figure.
12) Page 18, line 676, the author is recommended to provide a quantitative assessment of the improvements in cost and time consumption using the biosensors comparing to using conventional analytical methods.
13) Page 22, line 853, please include the full name of CMOS in the text.
14) The author may consider combining figure 20 and figure 21 as they are related results.
Author Response
- Page 2, line 8, more references are recommended for the two categories of
biosensor materials.
Authors’ response:
Amendments were made.
Figure 1, the author should explain the meaning of all the patterns (like the
bacteria-shaped pattern) drawn in the figure. The details should be included in the
figure description.
Authors’ response:
Further description was added in the figure description.
- Figure 2 may not be necessary to summarize the advantages of plasmonic and
metamaterial-based sensors. The author may summarize the benefits in a table with
references, or in the main text.
Authors’ response:
The figure was removed and replaced with description within the text and accordingly all the figure numbers have been changed in the captions and the in-text description.
- Page 3, line 80, please add references for surface plasmon resonance (SPR).
Authors’ response:
References were added.
- Page 4, line 114 , in the section for efficiency determination . The author is
encouraged to introduce how limit of detection (LOD) and specificity (s) are measured experimentally.
Authors’ response:
Further information was elaborated and a seubsection was added to give background to the experimental measurement of LOD.
- Page 5, line 152, please add reference for all the biological assays mentioned
here for detecting infectious viruses.
Authors’ response:
References were added for the detecting methods.
- Figure 4, the author should write the figure description in a detailed manner,
with brief introduction of all mentioned architectures.
Authors’ response:
Further details were added in the figure description.
- Page 9, line 301, when referring to a previous publication, in addition to the year
of publication, the author could also mention the name of the first author of that
publication.
Authors’ response:
Adjustment was made.
- Figure 7, please explain the workflow in figure description in a more detailed manner.
Authors’ response:
Further description was added.
- Figure 9, the author may want to rearrange the size of fig 9(b) and illustrate how
the scattering pattern is related to fig9(a).
Authors’ response:
Adjustments were made.
- Figure 12, the author should explain the meaning of the letters (h, t, d, w) used in the figure.
Authors’ response:
Description of the letters was added.
- Page 18, line 676, the author is recommended to provide a quantitative
assessment of the improvements in cost and time consumption using the biosensors
comparing to using conventional analytical methods.
Authors’ response:
Unable to get exact values for a quantitative assessment as they are dependent on multi factors. Hence, table 2 was providing an overall comparison between the various detection methods.
- Page 22, line 853, please include the full name of CMOS in the text.
Authors’ response:
Amendment was made.
- The author may consider combining figure 20 and figure 21 as they are related
results.
Authors’ response:
Figures were combined into one figure.
Reviewer 2 Report
The "Plasmonic Biosensor: Review" is an extensive collection of plasmonic biosensor works. In this paper the authors meticulously compared different plasmonic techniques in terms of design, fabrication, relative advantages and drawbacks. This is an extensive review and can be a very nice reference for new and existing researchers in the field. I recommend this for publication in the Biology journal. Here are two minor comments that should improve the quality of the presentation.
- the various fonts and labels are too small to read, perhaps this would be a better figure if the structures were made from scratch. The following figures need to be replaced with high resolution pictures. Also, where applicable, please refer to the papers that they have been taken from. Fig1, Fig3b, Fig4 (partially), Fig.6, 7, 8, 11, 15, 19.
- Fig2: The schematic to my understanding represents advantages of biosensors. What do the arrows represent ? It is confusing and seems the latter somehow follows from the former.
Author Response
- the various fonts and labels are too small to read, perhaps this would be a better figure if the structures were made from scratch. The following figures need to be replaced with high resolution pictures. Also, where applicable, please refer to the papers that they have been taken from. Fig1, Fig3b, Fig4 (partially), Fig.6, 7, 8, 11, 15, 19.
Authors’ response:
Images 6 ,11, 15, and 19 were replaced with clearer labels and higher resolution images.
Images 1, 3b, 4, 7, and 8 are made from scratch and hence were not referenced to a paper.
Fig2: The schematic to my understanding represents advantages of biosensors. What do the arrows represent ? It is confusing and seems the latter somehow follows from the former. Arrows were replaced by lines. There is no sequencing in the advantages. It just previews the various advantages of plasmonic bio-sensors. They are represented by a schematic diagram for ease of preview, rather than listing them down.
Authors’ response:
The figure was removed and replaced with description within the text and accordingly all the figure numbers have been changed in the captions and the in-text description.